# Latent Policy Barrier: Learning Robust Visuomotor Policies by Staying In-Distribution

**Zhanyi Sun**      **Shuran Song**
Stanford University
project-latentpolicybarrier.github.io

## Abstract

Visuomotor policies trained via behavior cloning are vulnerable to covariate shift, where small deviations from expert trajectories can compound into failure. Common strategies to mitigate this issue involve expanding the training distribution through human-in-the-loop corrections or synthetic data augmentation. However, these approaches are often labor-intensive, rely on strong task assumptions, or compromise the quality of imitation. We introduce Latent Policy Barrier, a framework for robust visuomotor policy learning. Inspired by Control Barrier Functions, LPB treats the latent embeddings of expert demonstrations as an implicit barrier separating safe, in-distribution states from unsafe, out-of-distribution (OOD) ones. Our approach decouples the role of precise expert imitation and OOD recovery into two separate modules: a base diffusion policy solely on expert data, and a dynamics model trained on both expert and suboptimal policy rollout data. At inference time, the dynamics model predicts future latent states and optimizes them to stay within the expert distribution. Both simulated and real-world experiments show that LPB improves both policy robustness and data efficiency, enabling reliable manipulation from limited expert data and without additional human correction or annotation.

## 1 Introduction

In control theory, Control Barrier Functions (CBFs) offer a principled mechanism to enforce safety in autonomous systems by explicitly defining a safety set and ensuring states remain within it during execution [2]. Motivated by the conceptual clarity of CBFs, we seek a similar mechanism for visuomotor policies learned purely from data, where explicit analytical definitions of safety constraints and system dynamics are typically unavailable. In learning-based visuomotor control, the analogous challenge arises from covariate shift [55]: minor deviations from demonstrated expert behaviors can quickly compound, pushing agents into out-of-distribution (OOD) states and causing task failures. Traditional behavior cloning (BC) methods are particularly vulnerable to covariate shift, severely limiting their effectiveness and reliability in real-world applications.

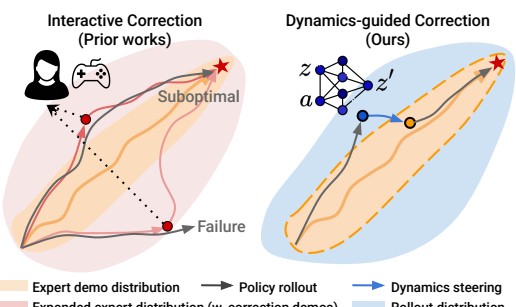

Figure 1: To mitigate covariate shift, prior DAgger-like methods (Left) expand the training distribution via human-in-the-loop corrections (—), often introducing suboptimal trajectories. In contrast, our **Latent Policy Barrier** treats the expert distribution (■) as an implicit *barrier*, using a learned dynamics model to *detect* deviations (●) and *steer* the policy back toward expert behavior (●). The dynamics model is trained on both expert demos (■) and policy rollouts (■), requiring no additional human input.

Current approaches to address covariate shift primarily involve **expanding the training distribution**. One common approach is human-in-the-loop interventions, such as DAgger [11, 55, 36, 29, 40, 39], which repeatedly collect corrective demonstrations whenever agents deviate from expert behaviors

39th Conference on Neural Information Processing Systems (NeurIPS 2025).

(Figure 1 left). Another strategy involves synthetic data augmentation methods that leverage task-specific invariances [17, 42]. However, these approaches are either labor-intensive or rely heavily on strong prior assumptions about the environment. Moreover, both approaches risk introducing inconsistent or suboptimal demonstrations into the training dataset, potentially degrading overall policy performance [41]. This limitation exposes a fundamental **trade-off** in robust visuomotor policy learning: precise imitation benefits from consistent, high-quality expert datasets, whereas robustness inherently demands exposure to diverse and often suboptimal data. Constructing a single dataset that balances these competing needs is inherently challenging.

To resolve this tension, our key insight is to explicitly **decouple** these conflicting objectives (Figure 1 right). Inspired by CBFs, we propose **Latent Policy Barrier (LPB)**, which implicitly treats the latent expert demonstration distribution itself as a barrier that separates safe, in-distribution states from unsafe, out-of-distribution regions [7]. Unlike analytically defined CBFs, LPB does not require explicit safety sets or system dynamics. Instead, it maps high-dimensional expert states into a learned latent space and uses their embeddings to define a barrier for detecting and correcting deviations.

To decouple precise expert imitation and OOD recovery, LPB leverages two complementary components: (a) a base diffusion policy trained exclusively on consistent, high-quality expert demonstrations, ensuring precise imitation; and (b) an action-conditioned visual latent dynamics model trained on a broader, mixed-quality dataset combining expert demonstrations and automatically generated rollout data [4, 45, 9, 62]. We collect the rollout data by executing intermediate checkpoints saved during base policy training. Importantly, the rollout data naturally covers diverse deviations around the policy's own distribution without requiring explicit success labels, task rewards, or additional human teleoperation. At inference time, LPB ensures that the agent stays within the expert distribution by performing policy steering in the latent space. LPB uses the dynamics model to predict future latent states conditioned on candidate actions output from the base policy. Then LPB minimizes the distance between the predicted future latent states and their nearest neighbors from the expert demonstrations in the same latent space. This latent-space steering approach simultaneously achieves high task performance and robustness, resolving deviations without compromising imitation precision.

In summary, we introduce Latent Policy Barrier in the context of behavior cloning. LPB offers the following advantages: 1) improves **sample efficiency** by decoupling expert imitation from out-of-distribution correction - enabling the policy to focus on learning from a small amount of high-quality human demonstrations; 2) enhances **robustness** through the use of a dynamics model trained on inexpensive, lower-quality policy rollout data; and 3) **plug-and-play compatibility** with off-the-shelf pre-trained policies, improving their robustness without requiring policy retraining or fine-tuning.

Experimental results across simulated and real-world manipulation tasks demonstrate that LPB is able to enhances both the robustness and sample efficiency of visuomotor policy learning. Code and data for reproducing the result will be made publicly available.

## 2  Related Work

**Mitigating Covariate Shift in Imitation Learning**: Behavior Cloning (BC), despite its simplicity, remains a strong baseline for working solely with expert demonstrations [51]. An extensive body of works mitigates covariate shift of BC by expanding the expert distribution. Prior works use interactive expert interventions [55, 36, 29, 40, 14, 39, 25] that are iteratively integrated into the policy learning loop to improve performance. Alternative ways to expand the training distribution include synthetic data generation with task invariances [17, 42, 65], noise injection [36, 27], or dynamics model-guided state-action pair generation [49, 28]. Instead of focusing on the data, Inverse Reinforcement Learning combats covariate shift by alternately collecting on-policy rollouts and updating reward function that penalizes deviations from expert trajectories, pulling the learner's state distribution toward the expert's [68, 24, 18]. Other lines of work include training a recovery policy that automatically pushes the agent back to the in-distribution region [53] and incorporating training objectives that penalize distribution divergence [43]. Our method differs from these strategies by performing inference-time steering in latent space, bypassing the need for additional expert queries or heavy data augmentation.

**Policy Learning from Suboptimal Data**: Offline RL algorithms prevents the learned policy from drifting into OOD states by imposing an explicit regularization. This can take the form of policy regularization that uses a penalty to keep the learned policy close to the behavior policy or a BC prior [60, 34, 20, 19, 50], conservative value estimation that pessimistically down-weights out-of-distribution actions [35, 33], or model-based uncertainty quantification that learns a dynamics

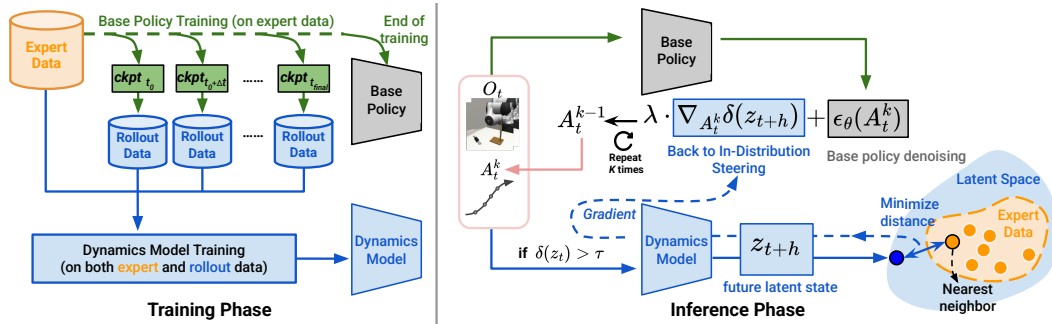

Figure 2: **Latent Policy Barrier**: 1) During **training**, we train a base diffusion policy on expert demonstration data and a visual latent dynamics model on both the expert and rollout data. 2) During **inference**, LPB keeps the agent within expert states by steering the base policy in the latent space. We define the $\ell_2$ distance of a latent state to its nearest expert latent state as the latent OOD score $\delta$. If $\delta(z_t)$ exceeds a predefined threshold, back to in-distribution steering is invoked. Details on the inference-time steering process can be found in Algorithm 1.

model and uses an uncertainty penalty to discourage trajectories from unfamiliar regions [64, 30, 8]. These strategies can exploit large, mixed-quality datasets, but they assume an external reward. Methods that learn from suboptimal demonstrations fit a reward function via enforcing trajectory rankings or regressing returns against noise-perturbed rollouts [5, 10]. Recent work shows that an agent can exploit its own rollouts - using them to forecast and refine future latent trajectories [62], to autonomously accumulate new task executions [4], or to label and filter heterogeneous demonstrations [9] - to supplement the limited expert data. In contrast, we repurpose policy rollout data to learn a dynamics model that, at inference time, nudges the agent back toward the expert distribution.

**Inference-Time Policy Steering:** A growing line of work improves a pre-trained policy at inference time by steering, without requiring additional data or fine-tuning. Guided denoising biases diffusion-based policies toward specified goals or reward signals by injecting gradient guidance into the denoising process [26, 1, 54, 46]. Value-based filtering samples actions from a generalist policy and executes the one ranked highest by a value function [47]. Human-in-the-loop steering treats user provided sub-goals, corrections, or preferences as constraints injected into the policy's sampler [58, 61]. Model-predictive refinement combines a base policy with a dynamics model to improve task performance or respect safety constraints [67, 56, 52, 48]. Similarly, LPB also uses a latent visual dynamics model [22, 21, 23, 59, 66] to steer the policy. Unlike prior works, LPB performs gradient-based action corrections using gradients from the latent dynamics model. The corrections pull predicted future states toward the expert manifold, unifying classifier guidance with model-predictive foresight while needing no explicit goals, rewards, or human input.

## 3 Method: Latent Policy Barrier

We consider the problem of visuomotor policy learning with behavior cloning. Given a demonstration dataset of observation-action pairs $\mathcal{D}_{\text{expert}} = \{(o_t, a_t)\}_t$, the goal of the base policy is to learn a $\pi : \mathcal{O} \to \mathcal{A}$ that maps observations to actions by imitating expert behavior. Following the notation introduced by [12], at each time step $t$, the policy $\pi_\theta(A_t|O_t)$ receives the most recent $T_o$ observations $O_t = \{o_{t-T_o+1}, \ldots, o_t\}$ and predicts the next $T_p$ actions $A_t = \{a_t, \ldots, a_{t+T_p-1}\}$. Among these predicted actions, the first $T_a$ steps, $\{a_t, \ldots, a_{t+T_a-1}\}$, are executed without replanning. The process then repeats after receiving new observations.

The goal of Latent Policy Barrier is to learn such a robust policy $\pi_\theta$ from a limited set of high-quality expert demonstrations, capable of consistently performing long-horizon robot manipulation tasks with minimal compounding errors and drifting into out-of-distribution states (Figure 2). In the following subsections, we detail our approach: base policy and dynamics model training §3.1, and test-time optimization to avoid deviations §3.2.

### 3.1 Policy and Dynamics Model Training

We train the policy $\pi_\theta(A_t|O_t)$ using a diffusion model [12] on a limited set of high-quality expert demonstrations. Given the high-dimensional visual observations, the policy uses a visual encoder

$h_\theta(O_t)$ that maps raw image observations into latent representations. The visual encoder and the noise-prediction network are trained end-to-end with behavior cloning loss.

To enhance dynamics model's generalization beyond the distribution of expert demonstrations, we collect an additional dataset of exploratory rollouts. These trajectories are automatically generated by rolling out intermediate checkpoints of the diffusion policy saved during training. Concretely, after an initial warm-up phase of $t_0$ training epochs, during which the base policy is still highly random, we save policy checkpoints at fixed intervals. Every $\Delta t$ epochs we save a checkpoint $\text{ckpt}_{t_0+n\Delta t}$ and roll it out for $N$ full episodes in the evaluation environment, recording all transitions regardless of task success or failure. This schedule continues until the final training epoch $t_{\text{final}}$. The resulting rollout dataset covers both early exploratory and near-convereged behavior, yielding a much wider state–action distribution than the original expert demonstrations. The diverse transitions in the rollout dataset enables the dynamics model to generalize more effectively, especially to novel states encountered at test time. Importantly, the rollout data is collected without human correction or reward annotations. Since it is generated as a natural byproduct of base policy training, no additional teleoperation or manual labeling is required. This makes it a inexpensive source of training data for learning a generalizable dynamics model that can steer the policy back toward expert-demonstrated states. We ablation the choice of data source for dynamics model training in Appendix A.

The visual latent dynamics model, denoted as $d_\phi$, is trained to predict future latent observations given current observations $O_t$ and a sequence of future candidate actions $A_t$. It consists of two components: a frozen visual encoder $h_\theta$, shared with the base policy, and a learnable dynamics predictor $f_\phi$. The full model is thus written as:

$$d_\phi(O_t, A_t) = f_\phi(h_\theta(O_t), A_t) \tag{1}$$

We reuse the visual encoder $h_\theta$ from the behavior cloning policy, which is trained end-to-end to optimize for task execution. This design choice is motivated by the following consideration: during test-time optimization, the dynamics model is used to predict future latent states in the same embedding space that the policy relies on to make action decisions. Freezing the encoder ensures consistency between the latent representations used by the policy and those optimized through the dynamics model (3.2). It also stabilizes training by preventing collapse of the representation space. The dynamics predictor $f_\phi$ is implemented as a decoder-only transformer ([57, 44, 6, 66]. Given encoded latent representations of current observations and a future action sequence, the dynamics model predicts the encoded future latent observation at a specified prediction horizon.

The model is trained using a latent-space mean squared error (MSE) loss between the predicted future latent and the ground-truth latent of the future observation:

$$\mathcal{L}_{\text{dynamics}}(O_t, A_t, o_{t+T_p}) = \|h_\theta(o_{t+T_p}) - f_\phi(h_\theta(O_t), A_t)\|_2^2 \tag{2}$$

### 3.2 Test-Time Optimization

At inference time, our goal is to mitigate compounding errors by steering the policy back toward in-distribution expert states whenever deviations from those states are about to occur. We detect OOD states using the latent OOD score, which measures how far the current observation lies from the expert data distribution in the latent space. Specifically, given an observation $o$, we first encode it into a latent representation $z = h_\theta(o)$ using the frozen visual encoder $h_\theta$ from the base policy. To quantify distributional shift, we identify the nearest neighbor of $z$ in the expert latent space, denoted as $z^{\text{NN}} \in D_{\text{expert}}$. The latent OOD score $\delta(z)$ is defined as:

$$\delta(z) = \|z - z^{\text{NN}}\|_2^2 \tag{3}$$

A higher $\delta$ value indicates that $z$ is farther from the expert data manifold and is thus more likely to be out-of-distribution for the base policy. At inference timestep $t$, we employ the learned dynamics model $d_\phi$ to refine the stochastic denoising process (Algorithm 1). At timestep $t$, the agent observes $O_t$, which is input to the base policy's denoising process. We first compute the latent OOD score for the current observation. If it's lower than a pre-defined threshold $\tau$, then the action generated by the base policy is executed. Otherwise, we refine the denoising process using gradient guidance. Taking inspirations from classifier guidance in diffusion models ([15]) and its applicatons in robotics-related domains ([1, 26, 63]), we take the gradient of latent OOD score of predicted future latent state through the dynamics model and use this gradient to refine the action denoising process. Denote the noisy action sample at timestep $t$ and at denoising iteration $k$ as $A_t^k$, we extend classifier guidance to refine

$A_t^k$ by minimizing the latent OOD score of predicted future state $z_{t+h} = d_\phi\big(h_\theta(O_t),\ A_t^k\big)$. We define the modified noise prediction as

$$\hat{\epsilon}\big(A_t^k\big) \;=\; \epsilon_\theta\big(A_t^k\big) \;-\; \eta\,\sqrt{1-\bar{\alpha}_k}\,\nabla_{A_t^k}\,\delta\Big(d_\phi\big(z_t,\ A_t^k\big)\Big) \tag{4}$$

where $\eta$ is the guidance scale. The classic classifier guidance approach requires training a classifier on noisy data samples, which is less practical in robotics settings where executing random actions can be unsafe or infeasible. Our rollout data offers a practical alternative: actions in policy rollouts naturally exhibit greater variability than expert demonstrations. As a result, the dynamics model trained on this data is well-equipped to predict the outcomes of noisy action samples. To avoid unreliable guidance from highly stochastic early denoising steps - where actions diverge significantly from the expert manifold - we restrict gradient-based guidance to only the final $K_{\text{guide}}$ denoising steps, where samples are more structured. For instance, Diffusion Policy often uses up to 100 denoising steps with DDPM, we only choose to refine only a subset of the denoising steps, e.g., the last 10 steps. By reweighting the action sampling distribution in favor of actions that lower the latent OOD score of predicted future states, the gradient guidance effectively steers the agent toward expert-like states. To better understand the effectiveness of the gradient guidance strategy, we also compare against alternative optimization methods, including direct gradient descent on the action sequence and model predictive control (MPC). Ablation results are provided in Appendix A.

---

**Algorithm 1** Latent Policy Barrier (Inference time)

---

**Require:** Base policy $\pi_\theta$, dynamics model $d_\phi$, visual encoder $h_\theta$, expert dataset $D_{\text{expert}}$, latent OOD score threshold $\tau$
  1: **Preprocess:** Encode all expert observations into latent states: $Z_{\text{expert}} = \{h_\theta(o) \mid o \in D_{\text{expert}}\}$
  2: **for** each timestep $t$ **do**
  3:     Observe $O_t = \{o_{t-T_o+1}, \ldots, o_t\}$
  4:     Encode latent state $z_t = h_\theta(o_t)$
  5:     Compute latent OOD score $\delta(z_t)$ by performing nearest neighbor search in $Z_{\text{expert}}$ (Eq. 3)
  6:     **if** $\delta(z_t) > \tau$ **then**                                    ▷ *Out-of-distribution detected*
  7:         **for** denoising step $k = K, \ldots, K - K_{\text{guide}}$ **do**
  8:             Sample noisy action $A_t^k$ from $\pi_\theta$
  9:             Predict future latent state: $\hat{z}_{t+h} = d_\phi(z_t, A_t^k)$
 10:             Compute gradient: $\nabla_{A_t^k}\delta(\hat{z}_{t+h})$
 11:             Apply gradient guidance to get $\hat{\epsilon}(A_t^k)$ (Eq. 4)
 12:         **end for**
 13:         Output final action sample $A_t$ after $K$ denoising steps
 14:     **else**
 15:         Output $A_t \sim \pi_\theta(\cdot | O_t)$
 16:     **end if**
 17:     Execute first $T_a$ steps of $A_t$
 18: **end for**

---

## 4 Experiments

Our evaluation focuses on two questions: 1) Does LPB improve sample efficiency and robustness of visuomotor policy learning compared to baseline methods? 2) When the agent drifts, can LPB detect the deviation at the right moment and steer the agent back to expert-like states? We benchmark across a suite of challenging robotic manipulation tasks in both simulated environments and on a real robot (Figure 3). In simulation, we benchmark on three suites: *Push-T* [12, 16], Robomimic [41], and the multi-task, language-conditioned *Libero10* [38]. For Robomimic, we select its three most challenging tasks, *Square*, *Tool Hang*, and *Transport*. For the real robot experiment, we test on the *Cup Arrangement* task from [13] and the *Belt Assembly* task from the NIST board assembly challenge [32].

### 4.1 Simulation Benchmarks

To evaluate the sample efficiency of our method, we focus on a limited demonstration regime, where suboptimal rollout data can play a significant role in improving performance. For each Robomimic task (*Square*, *Tool-Hang*, *Transport*) and for *Push-T*, we keep $20\%$ of the original expert demonstrations. For each task, a base diffusion policy is trained on these demonstrations. For *Libero10*, we use

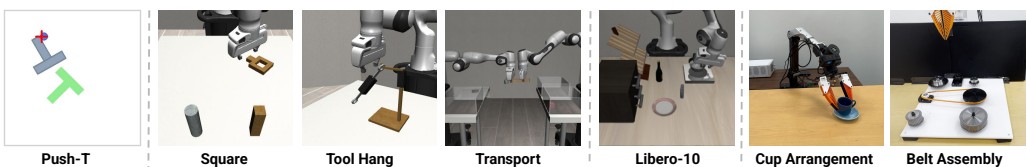

| Push-T | Square | Tool Hang | Transport | Libero-10 | Cup Arrangement | Belt Assembly |

Figure 3: **Benchmark Tasks.** We evaluate our method on a diverse set of manipulation tasks in both simulation and real-world.

| Task | Expert BC | Mixed BC | Filtered BC | CCIL | CQL | **Ours** |
|---|---|---|---|---|---|---|
| Square | $0.56 \pm 0.016$ | $0.50 \pm 0.025$ | $\mathbf{0.65} \pm 0.009$ | $0.63 \pm 0.025$ | $0.0 \pm 0.0$ | $\mathbf{0.65} \pm 0.019$ |
| Transport | $0.68 \pm 0.016$ | $0.60 \pm 0.016$ | $0.79 \pm 0.034$ | $0.69 \pm 0.074$ | $0.0 \pm 0.0$ | $\mathbf{0.85} \pm 0.009$ |
| Tool Hang | $0.27 \pm 0.009$ | $0.24 \pm 0.016$ | $0.29 \pm 0.009$ | $0.14 \pm 0.025$ | $0.0 \pm 0.0$ | $\mathbf{0.39} \pm 0.009$ |
| Push-T | $0.51 \pm 0.021$ | $0.47 \pm 0.026$ | $0.59 \pm 0.008$ | $0.48 \pm 0.034$ | $0.29 \pm 0.021$ | $\mathbf{0.65} \pm 0.012$ |
| Libero-10 | $0.65 \pm 0.009$ | $0.50 \pm 0.017$ | $0.71 \pm 0.006$ | $0.61 \pm 0.026$ | $0.0 \pm 0.0$ | $\mathbf{0.75} \pm 0.038$ |

Table 1: **Success rates** across simulated tasks with 20% demonstrations. Mean and std. deviation over 3 checkpoints and on 50 different held-out environment initial conditions.

all 50 provided demonstrations for each of the ten tasks to train a language-conditioned, multi-task base diffusion policy. During policy training, we save intermediate checkpoints at fixed intervals and use them to collect additional rollouts. See Appendix B for further implementation details.

We compare LPB against the following baselines:

- **Expert BC** [12]: Diffusion policy trained on only expert data.
- **Mixed BC**: Diffusion policy trained on both rollout data and expert data.
- **Filtered BC** [9, 45, 4]: We augment the expert data with successful rollout trajectories and re-train the diffusion policy on the aggregated dataset. We use the success criteria defined by the original benchmarks.
- **CQL** [35]: Conservative Q-Learning (CQL) is an offline RL algorithm that learns a value function that explicitly penalizes overestimation of unseen actions. CQL requires task rewards. For *Push-T*, we use the dense reward from [12]; for Robomimic and Libero10 tasks, we use the sparse reward, which gives 1 on success and 0 otherwise.
- **CCIL** [28]: CCIL enhances the robustness of behavior cloning by generating corrective data to augment the original expert data. Since the original method was proposed for tasks with low dimensional state space, we adapt it to tasks with image observations by encoding images with the frozen visual encoder in the base policy trained with behavior cloning loss.

As shown in Table 1, under the limited-demonstration setting, LPB matches or exceeds every baseline on all simulated tasks, showing strong sample efficiency. The largest gains appear on the long-horizon, precision-sensitive tasks, *Tool-Hang* and *Transport*, highlighting LPB's ability to correct minor action deviations that otherwise compound over time. On the most challenging *Tool-Hang* task, **Filtered BC** offers only marginal improvement, indicating that even "successful" rollouts still contain suboptimal or inconsistent actions that hurt BC. **CQL** obtains zero reward on Robomimic and Libero10 tasks, possibly due to the absence of a dense reward function, consistent with results reported in the Robomimic paper [41].

**Inference-time Robustness.** To investigate how well LPB mitigates covariate shift, we inject noise perturbations to the output action at inference time. At every time step, with probability $p \in \{0.0, 0.1, 0.2, 0.3, 0.4\}$, a Gaussian noise is added to the output action. We evaluate LPB alongside **Filtered BC**, **Expert BC**, and **CCIL** on the *Transport* task. Figure 4a shows that LPB maintains the highest success rate across all noise levels. While the baselines exhibit worsening performance as $p$ increases, LPB degrades the least, indicating robustness to inference-time perturbations.

**Sample Efficiency on Expert Demonstrations.** We next vary the fraction of expert demonstrations used to train the base policy on *Tool-Hang*, from 20% up to the full dataset, to see whether LPB continues to add value as more expert data becomes available. Figure 4b shows that LPB's advantage is largest in the low-data regime (up to 60% demos), the margin narrows as the dataset approaches 100%, reflecting performance saturation.

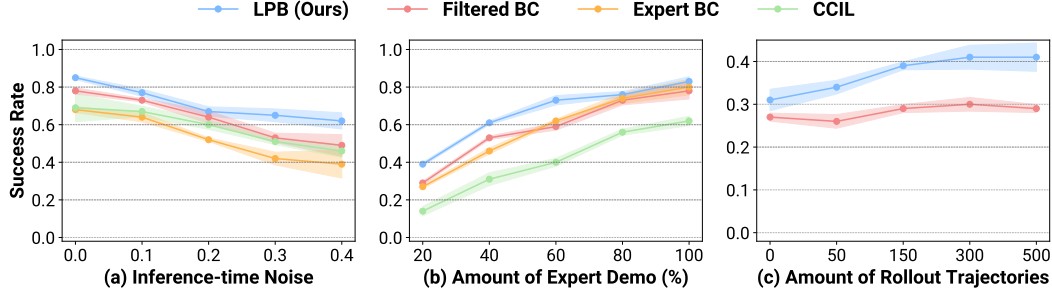

Figure 4: **(a)** Performance under perturbations on *Transport*, showing that LPB is more robust under external perturbation. **(b)** Impact of expert demonstrations on *Tool-Hang*, showing that LPB can achieve better performance under low-data regime (up to 60% demos). **(c)** Impact of rollout data on *Tool-Hang*, showing that LPB consistently improves performance as more rollout data is used to train the dynamics model despite the suboptimality in the data.

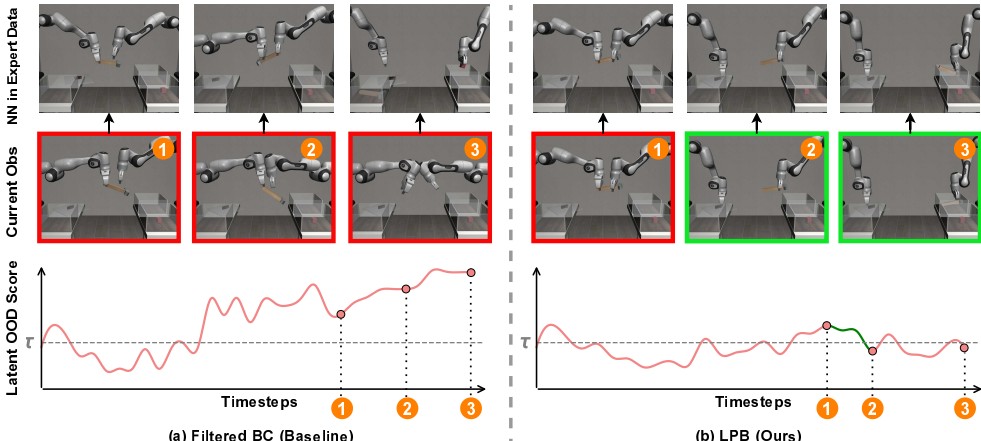

Figure 5: **Simulation Rollouts. Left**: **Filtered BC** baseline. **Right**: LPB. For both methods, we show RGB frames sampled from a representative rollout (middle row), the corresponding nearest expert demonstration state to the sampled frames (top row), and the latent OOD score $\delta$ over time (bottom row). The dashed line in the latent OOD score curve marks the threshold $\tau$ above which LPB's test-time optimization is invoked for. **Filtered BC** misses the two-arm hand-over, the latent OOD score spikes, and the episode fails. LPB ensures $\delta$ near $\tau$ throughout the entire rollout, stays on the expert manifold, and completes the task.

**Impact of Rollout Data Quantity.** To evaluate how the performance of LPB depends on the amount of rollout data used for dynamics model training, we conduct experiments on the *Tool Hang* task (Figure 4c). We vary the number of rollout trajectories used to train the dynamics model: 0, 50, 150, 300, and 500. When using 0 rollouts, the dynamics model is trained solely on expert demonstrations. We compare LPB against **Filtered BC**, which filters successful trajectories from the same rollout data and combines them with demonstrations to retrain a BC policy. Due to inconsistencies and suboptimal behaviors present even in the successful rollouts, Filtered BC yields little improvement upon the base policy. In contrast, LPB consistently improves as more rollout data is added, suggesting that the dynamics model benefits from the broader state-action coverage despite the suboptimality. Performance gains of LPB saturate around 300 rollouts.

**Latent OOD Score Visualization and Analysis.** To validate our design choice of the latent space and the latent OOD score formulation, we plot the latent OOD score over time for LPB and the **Filtered BC** baseline on a representative *Transport* rollout (Figure 5). As defined in 3.2, the latent OOD score is the $\ell_2$ distance between the encoded observation and its nearest expert neighbor, where the encoder comes from the base policy. In the rollout generated from the **Filtered BC** baseline, the two arms miss the hammer hand-over, leading to unrecoverable failure (frames with red bounding box). At these failure timesteps, the latent OOD score spikes above the threshold $\tau$, indicating the

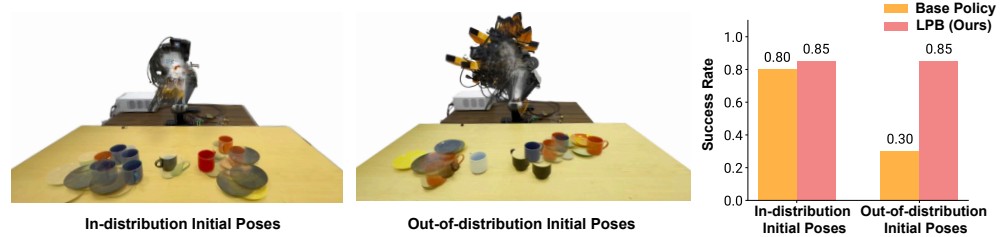

Figure 6: **Cup Arrangement Setup and Results.** Evaluation of the base policy (**Expert BC**) and **LPB** on the real-world *Cup Arrangement* task.

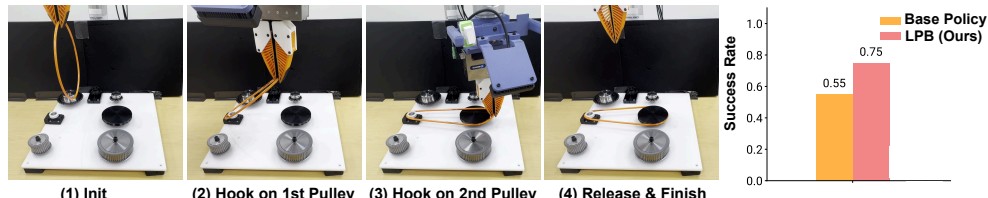

Figure 7: **Belt Assembly Setup and Results.** Evaluation of the base policy (**Expert BC**) and **LPB** on the real-world *Belt Assembly* task.

state has drifted far from the expert manifold. In the rollout generated by LPB the latent OOD score rises as the hand-over phase begins but stays near $\tau$; gradients from the dynamics model guides the policy back toward expert-like states (frames with green bounding box). Throughout the episode the score remains mostly below $\tau$, showing that LPB keeps the agent inside the demonstrated distribution. These observations support that (1) the visual encoder trained end-to-end with behavior cloning loss induces a latent space in which the nearest-expert distance reflects how far the current state has drifted from demonstrated behavior, and (2) this $\ell_2$ distance serves as a reliable signal for OOD detection and correction, enabling effective latent-space steering by LPB. See Appendix A for results on alternative latent space choices.

## 4.2 Real-World Evaluation

### 4.2.1 Cup Arrangement with an Off-the-Shelf Pretrained Policy

We first assess LPB's ability to improve the robustness of an off-the-shelf pretrained policy on a real robot. As the availability of such pretrained policies continues to grow [31, 3, 37], the **plug-and-play** nature of LPB and its ability to enhance their robustness makes it especially valuable.

**Task Setup.** Specifically, we use the pre-trained diffusion policy for the *Cup Arrangement* task as our base policy [13]. In this task, a robot arm with a wrist-mounted RGB camera and a compliant finray gripper needs to first rotate the cup so its handle faces right, then pick up the cup, and finally place the cup upright on the saucer.

**Base Policy and Dynamics Model.** The base diffusion policy checkpoint, trained on in-the-wild expert trajectories, serves as the base policy for LPB. To collect training data for the dynamics model, we roll out the pre-trained policy from deliberately out-of-distribution initial poses, gathering 80 trajectories. Because intermediate policy checkpoints are unavailable, we augment the dataset with additional 40 human play trajectories recorded via the handheld UMI device [13].

**Results.** We perform two groups of experiments: *in-distribution initial poses*, where the wrist camera initially observes both the cup and the saucer (Figure 6 left), and *out-of-distribution initial poses*, where the camera initially sees neither object (Figure 6 middle). As shown in Figure 6 right, LPB matches the base policy on in-distribution initial poses and substantially outperforms it on OOD initial poses, confirming that latent-space steering is able recover from distribution shifts without compromising nominal performance.

To illustrate how the dynamics model corrects distribution shift, Figure 8 plots four frames sampled from rollouts under the base policy (top row) and LPB (middle row), with LPB paired with its nearest expert demonstration image (bottom row) and the corresponding latent OOD scores. Both rollouts

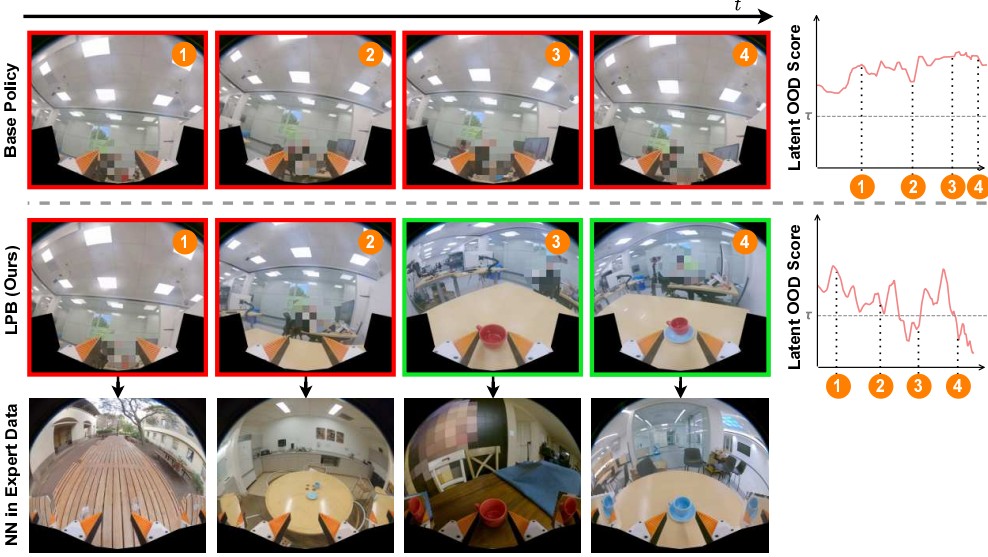

Figure 8: **Cup Arrangement Rollouts. Left:** RGB frames from base policy rollouts (top), LPB (middle), and the nearest expert demonstration to LPB (bottom). Red borders indicate out-of-distribution observations; green denotes in-distribution. **Right:** corresponding latent OOD scores. With LPB, the robot moves downward to reveal the tabletop and complete the task, while the base policy drifts further and reaches joint-limit.

start from the same wrist-camera pose showing neither cup nor saucer. Guided by the dynamics model, LPB tilts the camera downward to reveal the objects, driving its OOD score below the threshold and successfully completing the task. By contrast, the base policy continues moving in the wrong direction with its observations remain OOD throughout. It fails to locate the objects and terminates at a joint-limit failure. Although the demonstration data contains only in-the-wild images and the latent space is noisier than simulated tasks, the latent-space nearest neighbor selected by LPB still redirects the robot to in-distribution views, showing that LPB is robust to visual clutter and attends to task-relevant features.

#### 4.2.2 Belt Assembly

We further evaluate on the *Belt Assembly* task from the NIST task board.

**Task Setup.** As illustrated in Figure 7, the task begins with the belt already grasped by the robot's gripper. First, the robot positions the belt to hook it over the small pulley. Next, it moves downward while stretching the belt to loop its opposite side around the large pulley. The robot then performs a 180° rotation around the large pulley to thread the belt. Finally, it lifts the gripper to release the belt.

**Base Policy and Dynamics Model.** We collect 200 expert demonstrations, each performed with slight variations in the board position while keeping the initial robot pose fixed. To train the dynamics model, we additionally collect 400 rollout trajectories executed under randomly initialized board positions and initial robot poses.

**Results.** This task poses a significant challenge for naive behavior cloning due to its contact-rich nature and the high precision required for successful execution. At test time, we perform 40 rollouts for both the base policy and LPB, varying the initial robot pose and board position. All evaluations use the same set of test cases and initial robot configurations for fair comparison. The base BC policy achieves a success rate of 0.55. As shown in Figure 7 (right), LPB improves this performance to 0.75. Figure 9 shows representative rollout frames for both the base policy and LPB, along with their corresponding latent OOD score plots. At the start of the rollout, the observation lies out-of-distribution, reflected by a high latent OOD score. LPB first recovers the robot to an in-distribution pose, enabling the base policy to subsequently complete the task. In contrast, the base policy alone fails to recover, resulting in task failure.

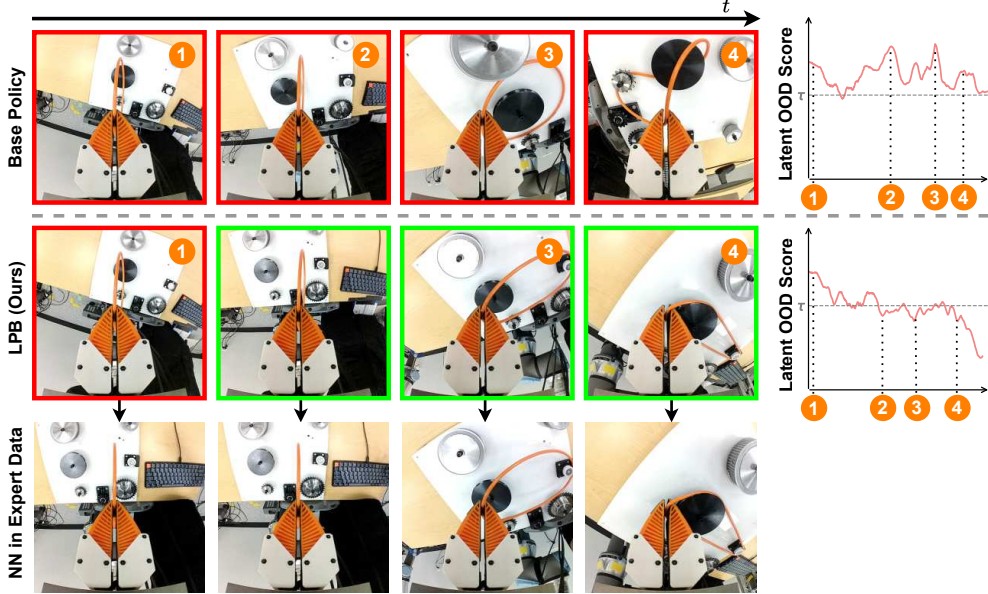

Figure 9: **Belt Assembly Rollouts. Left:** RGB frames from base policy rollouts (top), LPB (middle), and the nearest expert demonstration to LPB (bottom). Red borders indicate out-of-distribution observations; green denotes in-distribution. **Right:** corresponding latent OOD scores. LPB is able to help the robot recover to in-distribution expert states and complete the two threading motions.

## 5   Conclusion and Discussion

In this paper, we propose a novel method, Latent Policy Barrier, that transforms behavior cloning from a passive imitator into an active, self-correcting controller with a dynamics model. By treating the expert manifold as a latent barrier and steering a base policy with a learned dynamics model, LPB achieves reliable imitation and eliminates costly human corrections. Extensive simulated and real-robot experiments show that LPB surpasses or matches baselines when trained on just a small slice of the demonstration set, while maintaining robust performance under severe perturbations, highlighting a practical path toward reliable visuomotor learning.

**Limitations and Future Work.** Currently, LPB only corrects short-term, local deviations around the expert demonstration. Future work can expand the dynamics model training distribution and employ long-horizon reasoning to recover from more aggressive deviations. In addition, the visual latent dynamics model in LPB is trained per task, limiting reuse and requiring new data for each domain. Future work can explore training multitask dynamics models that capture shared dynamics structure, offering zero-shot generalization and amortized training cost across diverse robotics domains. Moreover, the latent OOD score in LPB assumes access to expert training data to define the in-distribution manifold; in scenarios where only a pretrained base policy is available without paired training data, this metric cannot be directly computed. Future work could investigate distribution-free uncertainty quantification to enable robust policy learning in the absence of expert training data.

## 6   Acknowledgements

We thank Mengda Xu for insightful discussions throughout the various stages of this project. We are also grateful to Yihuai Gao for assistance with the ARX arm setup and to Yifan Hou for support with the UR5 setup. We appreciate the valuable feedback from all REALab members on the manuscript. This work was supported in part by the NSF Awards #2143601, #2037101, and #2132519, and by the Toyota Research Institute. We thank TRI for providing the UR5 robot hardware and ARX for providing the ARX robot hardware. The views and conclusions contained herein are those of the authors and should not be interpreted as necessarily representing the official policies, either expressed or implied, of the sponsors.

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

# A   Additional Experiments and Results

## A.1   Ablation: Gradient Guidance Hyperparameter Choice

To understand how the gradient guidance hyperparameters in LPB affect the final task success rate, we run a set of experiments on the *Transport* task. We focus on the two hyperparameters: the guidance scale $\eta$ and the number of denoising steps that receive gradient guidance, $K_{\text{guide}}$. As shown in Figure 10 left, when $\eta$ is very small, the improvement over the base policy is minimal because the guidance signal is too weak. As $\eta$ increases, the improvement becomes more pronounced; however, once $\eta$ reaches 0.3, performance drops slightly, indicating an overshooting effect caused by overly strong guidance. Figure 10 right shows that LPB is generally robust to the choice of $K_{\text{guide}}$: even when gradients are applied only during the last 35 denoising steps, LPB still outperforms the base policy, demonstrating that the dynamics model can provide meaningful guidance even when the noisy action samples are far from valid expert actions. Across all tasks, we find that injecting gradient guidance during the final 10 denoising steps is sufficient for a substantial performance boost.

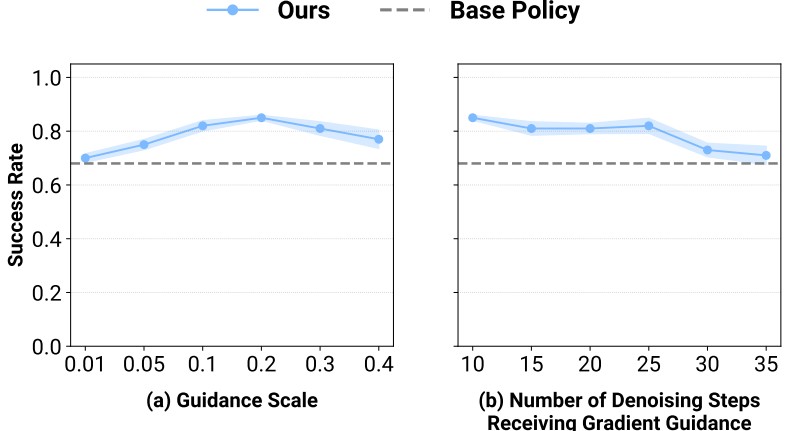

Figure 10: **(a)** Performance under varying guidance scale. **(b)** Performance under varying numbers of denoising steps that receive gradient guidance.

## A.2   Ablation: OOD Threshold

We analyze the effect of varying the OOD threshold for the *Transport* and *Tool-Hang* tasks. From Figure 11, we observe that setting too high results in fewer corrections being triggered, leading to performance drops. Conversely, setting too low can cause the dynamics model to get stuck in local optima and stall task progress. This effect is particularly pronounced in the *Tool-Hang* task, where the robot may get stuck at inserting the hook into the base.

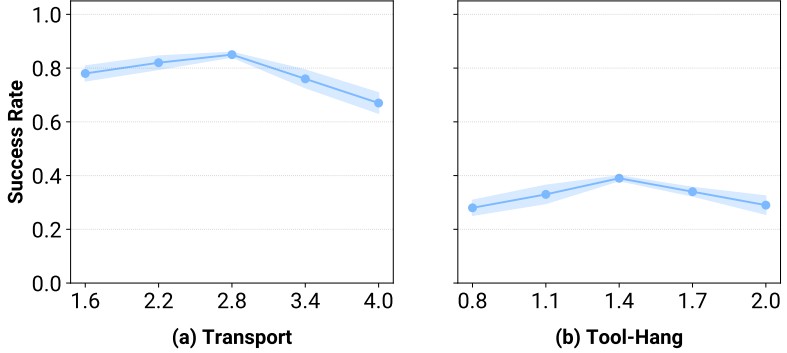

Figure 11: Performance under varying OOD thresholds.

## A.3   Ablation: Data Source for Dynamics Model Training

We evaluate different data sources for training the dynamics model on the *Tool-Hang* and *Transport* task. LPB leverages policy rollout data as an inexpensive and scalable way to improve the dynamics

model's generalization, but alternative sources are also available. We compare against two such baselines: **(1) Expert Demonstration with Noise Injection.** Gaussian noise is injected into expert demonstrations with a probability of 0.3 to generate diverse, perturbed trajectories. **(2) Epsilon-Greedy Exploration.** A base policy is first trained on limited expert data, then rolled out from randomized initial configurations with Gaussian noise added to the output action at a probability of 0.3 for exploration. As shown in Table 2, LPB, which relies on policy rollouts, achieves the highest success rate. We also note that rollout collection does not require manual tuning of noise magnitudes or injection probabilities, which simplifies the data collection procedure in practice. Moreover, it naturally captures a broad spectrum of data, spanning early exploratory trajectories to those generated by a converged policy.

Table 2: Success Rate with different data sources for dynamics model training

|  | Policy Rollout | Noisy Demos | Epsilon-Greedy |
|---|---|---|---|
| Transport | $\mathbf{0.85} \pm 0.009$ | $0.73 \pm 0.025$ | $0.71 \pm 0.024$ |
| Tool-Hang | $\mathbf{0.39} \pm 0.009$ | $0.30 \pm 0.031$ | $0.30 \pm 0.028$ |

## A.4 Ablation: Action Optimization

We ablate the action optimization strategies used in LPB on the *Tool-Hang* and *Transport* tasks. LPB adopts a classifier guidance-style approach for test-time action refinement. We compare this with two alternatives: **(1) LPB-MPC**, which uses Model Predictive Control with stochastic action sampling, and **(2) LPB-GD**, which directly optimizes actions via gradient descent. As shown in Table 3, LPB achieves the highest success rate. LPB-MPC also improves over the base policy, but not as much as LPB - likely because it relies on sampling, which can fail to capture rare but optimal actions if they are underrepresented in the action distribution. By contrast, LPB nudges each sampled action toward expert-like directions using a differentiable guidance signal, enabling exploration that remains on-manifold while still biasing toward high-value corrections. LPB-GD performs the worst, as its unconstrained optimization can produce actions that are out-of-distribution for both the base policy and the dynamics model.

Table 3: Success Rate for different action optimization strategies

|  | LPB (Ours) | LPB-MPC | LPB-GD |
|---|---|---|---|
| Tool-Hang | $\mathbf{0.39} \pm 0.009$ | $0.32 \pm 0.028$ | $0.26 \pm 0.028$ |
| Transport | $\mathbf{0.85} \pm 0.009$ | $0.80 \pm 0.043$ | $0.73 \pm 0.025$ |

## A.5 Ablation: Choice of Latent Representation

To study how the choice of latent space affects deviation detection and recovery toward expert states, we evaluate different latent representations on the *Square* and *Transport* tasks. We compare LPB, which uses the base policy's encoder, against two alternatives: **(1) DINOv2 Encoder** and **(2) Encoder Trained with Reconstruction**. For (1), we use a pretrained DINOv2 model as the observation encoder for the dynamics model; it outputs 256 patch embeddings of dimension 384. This representation preserves fine-grained visual details, but may include task-irrelevant information, as it is not directly trained with task supervision. For (2), we pretrain a ResNet-18 as an autoencoder using a reconstruction loss and then freeze it to serve as the encoder for dynamics model training. As shown in Table 4, both the base policy's encoder and DINOv2 improve over the base policy alone, with the base policy's encoder outperforming DINOv2. This may be because DINOv2 encoder, trained in a task-agnostic manner, may focus on irrelevant features, while the base policy's encoder, trained via behavior cloning loss, is more aligned with task-relevant information. The reconstruction-based encoder performs the worst, suggesting that the reconstruction objective alone does not produce a latent space suitable for effective action optimization.

Table 4: Success Rate with different latent space

| | Base Policy's Encoder | DINOv2 | Encoder Trained with Reconstruction |
|---|---|---|---|
| Square | $\mathbf{0.65} \pm 0.019$ | $0.61 \pm 0.025$ | $0.47 \pm 0.034$ |
| Transport | $\mathbf{0.85} \pm 0.009$ | $0.79 \pm 0.025$ | $0.68 \pm 0.043$ |

# B  Implementation Details

## B.1  Implementation Details of Latent Policy Barrier

**Base Policy.** We adopt Diffusion Policy as the base policy in LPB due to its strong capability in modeling complex robot action distributions. It uses a ResNet-18 as the image encoder and a U-Net as the noise prediction network, with FiLM layers to condition the denoising process on both observation features and the current diffusion timestep. For all tasks, the base policy is trained using 20% of the available expert demonstrations. Task-specific and shared hyperparameters are provided in Table 5 and Table 6, respectively.

**Dynamics Model.** Once base-policy training passes an initial warm-up of $t_0$ epochs, during which the policy remains highly exploratory, we begin saving checkpoints at fixed intervals. Concretely, every $\Delta t$ epochs we store $\text{ckpt}_{t_0+n\Delta t}$ and roll it out for $N$ complete episodes from randomly initialized configurations. All transitions, whether successful or not, are retained to train the dynamics model. This procedure continues until the final epoch $t_{\text{final}}$. The values of $t_0$, $\Delta t$, $t_{\text{final}}$, $N$, and the resulting total number of rollout trajectories for each simulated task are summarized in Table 7.

The dynamics predictor $f_\phi$ is implemented as a Vision Transformer (ViT). It receives the concatenation of the latent observation token, the proprioception state token, and an action token, and predicts the next latent observation and proprioception state tokens. Training hyperparameters for $f_\phi$ are provided in Table 8.

Table 5: Simulation task-dependent hyperparameters for base diffusion policy training.

| Env Name | $T_a$ | $T_p$ | #Demo | Training Epochs |
|---|---|---|---|---|
| Push-T | 8 | 16 | 41 | 500 |
| Square | 8 | 16 | 40 | 600 |
| Tool-Hang | 15 | 32 | 40 | 300 |
| Transport | 15 | 32 | 40 | 300 |
| Libero10 | 15 | 32 | 50 | 200 |

Table 6: Shared hyperparameters for base diffusion policy training.

| Name | Value |
|---|---|
| $T_o$ | 2 |
| Image Size | 140 |
| Crop Size | 128 |
| Batch Size | 64 |
| Learning Rate | $1 \times 10^{-4}$ |
| Diffusion Step | 100 |

Table 7: Rollout trajectories collection schedule for simulation tasks

| Env Name | $t_0$ | $\Delta t$ | $t_{\text{final}}$ | $N$ | Total |
|---|---|---|---|---|---|
| Push-T | 150 | 40 | 470 | 30 | 270 |
| Square | 70 | 50 | 470 | 30 | 270 |
| Tool-Hang | 70 | 50 | 270 | 30 | 150 |
| Transport | 70 | 50 | 270 | 30 | 150 |
| Libero10 | 40 | 40 | 160 | 50 | 200 |

Table 8: Hyperparameters for dynamics model training.

| Name | Value |
|---|---|
| History Length | 1 |
| Depth | 6 |
| Heads | 16 |
| MLP Dim | 2048 |
| Dropout | 0.1 |
| Batch Size | 64 |
| Learning Rate | $5 \times 10^{-4}$ |
| Training Epoch | 100 |

**Test-time Optimization.** At test time, we denoise actions with a DDPM scheduler. For each timestep $t$ we first compute the latent OOD score; if the score is below the threshold $\tau$, we run standard denoising and execute the resulting action $A_t^0$. If the score exceeds $\tau$, we apply latent steering during the final 10 denoising steps. Specifically, the current observation and the intermediate noisy action samples at denoising timestep $k$, $A_t^k$, are fed to the dynamics model, which predicts the future latent state $z_{t+h}$. We then measure the Euclidean distance between $z_{t+h}$ and its nearest expert state in latent space; this distance is back-propagated through the dynamics model, and the resulting gradient with

respect to the noisy action samples provides the guidance signal. The OOD threshold $\tau$ is chosen empirically by rolling out the final policy checkpoint, while the guidance scale $\eta$ is selected via a grid search. Both $\eta$ and $\tau$ for each task are listed in Table 9.

Table 9: Hyperparameters for test-time optimization

|  | $\eta$ | $\tau$ |
|---|---|---|
| Push-T | 0.05 | 3.2 |
| Square | 0.05 | 5 |
| Tool-Hang | 0.05 | 1.4 |
| Transport | 0.2 | 2.8 |
| Libero10 | 0.2 | 1.1 |

**Compute Resources.** All simulated experiments are run on a single NVIDIA L40S GPU (46 GB VRAM). Base policies require roughly 24–48 h to converge, while the dynamics models converge within 24 h. For each simulated task, we evaluate the final three checkpoints, each spaced 10 training epochs apart; the results reported in Table 1 are averages over those three checkpoints. In the real-robot setting, the base policy is pretrained and thus incurs no additional training cost. The dynamics model is trained in parallel on six NVIDIA L40S GPUs and converges in approximately 36 h.

## B.2    Implementation Details of Baselines

**Filtered BC.** This baseline uses the same rollout data collected for LPB. We discard trajectories that do not meet the task-specific success criteria and merge the remaining successful rollouts with the original expert demonstrations. The diffusion policy is then retrained from scratch on this augmented dataset.

**CQL.** We adopt the CQL implementation provided in the Robomimic codebase without modification.

**CCIL.** CCIL was originally designed for tasks with low-dimensional state inputs. To extend it to high-dimensional image observations, we cache latent representations produced by the base policy's encoder. Specifically, we first train a base diffusion policy $\pi_\theta$ on the limited expert demonstrations, identical to the setup used for LPB. We then encode every image observation in the expert dataset with $\pi_\theta$'s encoder and store the resulting latent transition pairs. CCIL's dynamics model is trained on these low-dimensional latents, after which CCIL's corrective-label generation procedure is applied to augment the expert data. Finally, we fine-tune $\pi_\theta$ on the augmented latent dataset. We use the authors' original implementation for all CCIL components.

## B.3    Real Robot Experiment Setup

### B.3.1    Cup Arrangement

**Setup.** We use a 6-DoF ARX5 robot arm with 3D-printed soft compliant fingers and a wrist-mounted GoPro camera, mirroring the sensor configuration of the original training data. Low-level controller is provided by the Universal Manipulation Interface (UMI) stack ported to the ARX5 platform https://github.com/real-stanford/umi-arx. The cup arrangement task consists of three sequential stages: (i) rotate the cup so its handle points right, (ii) grasp and lift the cup, and (iii) place it upright on a saucer.

**Pre-trained Base Policy.** We start from the publicly released diffusion policy checkpoint in the Universal Manipulation Interface repository https://github.com/real-stanford/universal_manipulation_interface. The base policy was trained on $1447$ in-the-wild demonstrations of the cup arrangement task collected in diverse environments. The test environment is not included in the training data. At each timestep, the policy receives a single RGB observation plus the past two proprioceptive states and predicts a $16$-step action sequence; the first $12$ actions are executed each control cycle.

**Dynamics Model Training.** To collect training data for the dynamics model, we roll out the pretrained policy from deliberately out-of-distribution initial poses, gathering $80$ trajectories. Because intermediate policy checkpoints are unavailable, we augment the dataset with additional $40$ human play trajectories recorded via the handheld UMI device. These play trajectories are intended to broaden state–action coverage rather than solve the task, for example, by sweeping the gripper

through random motions to capture wide-angle views. These 120 unlabeled trajectories, together with the original demonstrations, are used to train the dynamics model.

**Evaluation Protocol.** We evaluate two initial-pose regimes: (i) *in-distribution initial poses* and (ii) *out-of-distribution initial poses*. For each group we conduct 20 trials with both the base policy and LPB, using identical initial robot and object poses for fair comparison. During deployment we denoise actions with a DDIM scheduler (16 diffusion steps) and apply gradient guidance during the final five steps.

### B.3.2    Belt Assembly

**Setup.** We use a 6-DoF UR5 robot arm with 3D-printed soft compliant fingers and a wrist-mounted OAK-D camera. Low-level controller is provided by https://github.com/yifan-hou/hardware_interfaces.

**Base Policy Training.** We collect 200 expert demonstrations, each performed with slight variations in the board position while keeping the initial robot pose fixed. We train the base policy for 800 epochs. At each timestep, the policy receives a single RGB observation plus the past two proprioceptive states and predicts a 32-step action sequence; the first 16 actions are executed each control cycle.

**Dynamics Model Training.** To train the dynamics model, we collect 400 rollout trajectories executed under randomly initialized board positions and initial robot poses. We use hyperparameters $t_0 = 200$ $\Delta t = 200$, $t_{\text{final}} = 800$, and $N = 100$. We train the dynamics model with the combined 600 trajectories.

**Evaluation Protocol.** We perform 40 rollouts for both the base policy and LPB, varying the initial robot pose and board position. All evaluations use the same initial robot and object poses for fair comparison. During deployment we denoise actions with a DDIM scheduler (16 diffusion steps) and apply gradient guidance during the final five steps.

## C    Broader Impact

Our work contributes to improving the robustness of visuomotor policies in robotic manipulation settings by introducing a test-time optimization framework that leverages learned dynamics and latent-space guidance. This has the potential to reduce failure rates in deployment, especially in out-of-distribution scenarios, which is critical for the safe and reliable operation of robots in human-centered environments. However, as with any data-driven system, care must be taken to ensure that failure modes are thoroughly evaluated before real-world deployment. Future extensions could investigate formal guarantees for test-time optimization mechanisms.

