# OpenReview forum: "Latent Policy Barrier: Learning Robust Visuomotor Policies by Staying In-Distribution"
_NeurIPS.cc/2025/Conference — NeurIPS 2025 spotlight_

### Official Review · Reviewer_hbP6 · 2025-06-30

**Clarity:** 3
**Significance:** 3
**Originality:** 3
**Rating:** 5
**Confidence:** 4

**Summary:**

This paper introduces an approach to keep diffusion policies in distribution during execution. The core idea is to learn a local dynamics model by gathering data using policy rollouts from a trained policy (or checkpoints from the policy during training). The trained dynamics model is then used as a form of classifier guidance to keep the observed latents close to a set of latent states gathered from an expert's demonstration. This helps keep the policy close to the support of the expert training data, and improves robustness. Results show this reduces the amount of data required for robust model training. Experiments are conducted in simulation and on real robots.

**Questions:**

In Fig 4c. How is the impact of rollout data computed and contribution to success rates? Eg. Where does the 0.4 number come from at 300 trajectories.

I may have missed this, but what sort of coverage do you need for the expert latents? How big is the buffer, and do you do anything special to ensure they adequately cover the space of interest?

**Ethical Concerns:**

["NO or VERY MINOR ethics concerns only"]

**Final Justification:**

The authors included additional experiments that strengthen and already clear paper. I think this should be accepted.

**Limitations:**

Yes.

**Quality:**

3

**Strengths And Weaknesses:**

Strengths:

The idea is simple, and results show that it is robust to noise, and policies require fewer expert demonstrations to be effective.

I think it is a sensible approach, and like that it is modular and can be attached to a pre-trained policy.

This meets my internal criteria of would I send it to a PhD student.

The paper is well written, and experiments are nicely executed.

Weaknesses:

The data collection process for dynamics means that only a small local model can be trained, this means that the approach may struggle in complex environments with greater environmental dynamics and for longer horizon tasks. This is discussed in the limitations.

Many of the demonstration tasks show the approach successfully driving the robot to the expert data distribution in areas like free space motion, but I would have liked to see an example of this happening in a more complex interactive setting like contact, where classical planning techniques couldn't work.

This is nitpicking, but the distance used to determine whether something is out of distribution is an l2 distance between the nearest expert latent. Given that the policy being corrected is a probabilistic one, it seems that a less heuristic approach would be to directly determine where a latent lies in distribution or not, although this may require solving the flow equation.

---

> ### Author Rebuttal · Authors · 2025-07-31
>
> We thank Reviewer hbP6 for their time and expertise in reviewing our submission. Their insightful comments are valuable in helping us improve the quality of our work.
> > *“Many of the demonstration tasks show the approach successfully driving the robot to the expert data distribution in areas like free space motion, but I would have liked to see an example of this happening in a more complex interactive setting like contact, where classical planning techniques couldn't work.”*
>
> We thank the reviewer for this insightful question. To demonstrate LPB’s effectiveness in contact-rich settings, we evaluate it on an additional simulation benchmark and a real-world task.
>
> In simulation, we evaluate on Libero-10 [1], a multi-task, language-conditioned manipulation benchmark that contains 10 different tasks. For each task, Libero-10 provides 50 expert demonstration trajectories. We use those $50\times 10$ expert trajectories to train the base policy. For each task, we collect 200 rollout trajectories. We use $200\times 10$ rollout trajectories and the 500 expert trajectories to train the dynamics model. Both the base policy and the dynamics model are multi-task with an additional shared language encoder. We perform 50 rollouts per task across ten tasks and report the average success rate.
>
> | Method        | Libero-10 Success Rate |
> |---------------|------------------------|
> | Expert BC     | 0.65 ± 0.009           |
> | Mixed BC      | 0.50 ± 0.017           |
> | Filtered BC   | 0.71 ± 0.006           |
> | CCIL          | 0.61 ± 0.026           |
> | Ours (LPB)    | **0.75 ± 0.038**       |
>
> For the real-world evaluation, we consider the **belt assembly task** from the NIST task board, which involves hooking a rubber band onto two pulleys, a contact-intensive manipulation task. We collect 200 trajectories to train the base policy and 400 rollout trajectories to train the dynamics model. We conduct 40 rollouts using both the base policy and LPB. LPB boosts the success rate from **0.55 to 0.75**. Please refer to the original project website for rollout videos.
>
> > *“This is nitpicking, but the distance used to determine whether something is out of distribution is an l2 distance between the nearest expert latent. Given that the policy being corrected is a probabilistic one, it seems that a less heuristic approach would be to directly determine where a latent lies in distribution or not, although this may require solving the flow equation.”*
>
> We thank the reviewer for the insightful question. We agree that our use of L2 distance to the nearest expert latent is a simple heuristic. While effective in practice, it does not fully leverage the structure of the underlying diffusion model.
>
> We fully agree that more principled approaches to uncertainty quantification that are grounded in the generative structure of diffusion or flow-based models could offer a richer and more theoretically motivated signal. Prior work has proposed various OOD detection methods specifically tailored to generative models. Interestingly, [2] highlights that likelihood alone is not a reliable indicator for OOD detection. For diffusion models in particular, recent approaches include: using reconstruction consistency across denoising steps [3], modeling the distribution over predicted diffusion timesteps [4], using curvature along the diffusion paths [5], and leveraging approximate score matching loss computed at test time [6]. While these methods can be more computationally involved, they hold strong potential for improving both the interpretability and robustness of online guidance in frameworks like ours.
>
> We consider this an exciting direction for future work and appreciate the reviewer’s suggestion. **We will include this discussion in the final paper.**
>
> > *“In Fig 4c. How is the impact of rollout data computed and contribution to success rates? Eg. Where does the 0.4 number come from at 300 trajectories?”*
>
> We thank the reviewer for raising this question. In Fig. 4c, we evaluate the same base policy while varying the amount of rollout data used to train the dynamics model. The curve shows how the success rate of LPB-guided rollouts improves as the dynamics model becomes more accurate and generalizable with more rollout data. The success rate at each point is computed by averaging the task success across 50 randomized test configurations and random seeds. A success rate of 0.4 simply means 20 out of 50 rollouts are successful.
>
> The improvement with increasing rollout data reflects better coverage of the state-action space, which enables more reliable gradient-based corrections. However, we observe that the benefit plateaus around 300 trajectories, likely because the additional rollouts no longer significantly expand the distribution of reachable states encountered at test time. **We will make this more clear in the final paper**.
>
> > *“What sort of coverage do you need for the expert latents? How big is the buffer, and do you do anything special to ensure they adequately cover the space of interest?”*
>
> We thank the reviewer for this insightful question. Our method does not assume any explicit coverage requirement over the expert latents. The key requirement is that the expert demonstrations are optimal, high-quality, and consistent, so that the base policy trained on them performs precise imitation within the expert distribution. As shown in Fig. 4b, 100% demonstration corresponds to 200 trajectories. We found that even a modestly sized expert dataset (e.g., 20% of the data, or 40 expert trajectories) is sufficient to enable effective online correction with LPB.
>
> ### References
>
> [1] Liu, Bo, et al. *Libero: Benchmarking knowledge transfer for lifelong robot learning.* Advances in Neural Information Processing Systems 36 (2023): 44776-44791.
>
> [2] Nalisnick, Eric, et al. *Do deep generative models know what they don't know?.* arXiv preprint arXiv:1810.09136 (2018).
>
> [3] Graham, Mark S., et al. *Denoising diffusion models for out-of-distribution detection.* Proceedings of the IEEE/CVF Conference on Computer Vision and Pattern Recognition. 2023.
>
> [4] Livernoche, Victor, et al. *On diffusion modeling for anomaly detection.* arXiv preprint arXiv:2305.18593 (2023).
>
> [5] Heng, Alvin, and Harold Soh. *Out-of-distribution detection with a single unconditional diffusion model.* Advances in Neural Information Processing Systems 37 (2024): 43952-43974.
>
> [6] Lee, Sung-Wook, Xuhui Kang, and Yen-Ling Kuo. *Diff-dagger: Uncertainty estimation with diffusion policy for robotic manipulation.* arXiv preprint arXiv:2410.14868 (2024).

---

> > ### Comment · Reviewer_hbP6 · 2025-08-01
> >
> > Thank you for these additional experiments showing performance in more challenging tasks. I think this is a good paper that should be accepted.

---

### Official Review · Reviewer_jkE6 · 2025-07-01

**Clarity:** 3
**Significance:** 3
**Originality:** 3
**Rating:** 4
**Confidence:** 3

**Summary:**

This paper aims to address the covariate shift problem that is prevalent in visuomotor policies trained by behavior cloning (BC). The authors propose a new framework called Latent Policy Barrier (LPB). The core idea is to regard the embedding distribution of expert demonstrations in the latent space as an implicit "safe" region, and inspired by the control barrier function (CBF), ensure that the agent does not deviate from this region during policy execution. A key design of this framework is to decouple the two core goals in imitation learning - accurate imitation and out-of-distribution (OOD) recovery. It uses a base policy (the diffusion model in this paper) trained only on high-quality expert data to ensure the accuracy of imitation, and uses a dynamics model trained on expert data and suboptimal policy rollout data to learn how to recover from deviations. In the inference stage, the dynamics model "guides" the actions of the base policy through online optimization so that its predicted future latent states remain within the manifold of the expert data distribution, thereby improving the robustness of the policy without sacrificing the quality of imitation. Experiments on simulation and real robots demonstrate the effectiveness of the proposed method.

**Questions:**

On the quality of the latent space: The effectiveness of the entire LPB framework relies on a high-quality latent space. Can you discuss under what circumstances the visual encoder learned by the BC strategy may not form a "good" latent space (i.e., a space in which distances do not effectively reflect the degree of OOD)? For example, how does the performance of LPB change in the presence of a large amount of visual distractors that are irrelevant to the task?

On the sensitivity of hyperparameters: The setting of the threshold $\tau is crucial for the algorithm to trigger. How should this value be set in practice? Is it a sensitive parameter that needs to be fine-tuned for each task, or is it robust over a wide range? Can you provide more sensitivity analysis on \tau and the guidance strength \eta$?

Comparison with explicit recovery strategies: Your method can be viewed as an online, implicit recovery strategy. Another idea is to explicitly train a separate recovery strategy using the same rollout data. Can you fundamentally analyze the advantages and disadvantages of your proposed "online guidance" method compared with the "offline training recovery strategy" method and what are the applicable scenarios?

**Ethical Concerns:**

["NO or VERY MINOR ethics concerns only"]

**Final Justification:**

my decision is keeping my score

**Limitations:**

Yes. The authors clearly discuss the limitations of the current work in the conclusion section, including that the current method mainly deals with local deviations, the dynamics model is task-specific, and the method requires access to expert demonstration data to define the inner manifold of the distribution. The discussion is sufficient and honest.

**Quality:**

3

**Strengths And Weaknesses:**

Strengths:
The core idea of ​​the paper cleverly generalizes and applies the concept of "control barrier function" (CBF) in control theory to data-driven imitation learning, and proposes to use "latent space distribution of expert data" as an implicit barrier, which is a novel and insightful analogy and framework.

The paper contains in-depth qualitative and quantitative analysis. For example, the analysis in Figure 5 associates the latent space OOD score with the actual task failure moment, providing strong intuitive evidence for the core mechanism of the LPB method. Its ability to improve the "out-of-the-box" pre-trained model is verified on a real robot, which enhances the credibility and practical value of the paper.

Weaknesses:

Strictly speaking, the various components of the algorithm (diffusion strategy, latent space dynamics model, gradient-based guidance) are all existing technologies. The originality of this work is mainly reflected in the combination of these components and the guiding ideas behind them (CBF analogy and decoupling principle).

The method introduces some hyperparameters that need to be adjusted, such as the threshold $\tau of the latent space OOD score and the guidance strength \eta$. The sensitivity analysis of these hyperparameters is not sufficient in the main text of the paper, which may affect its reproducibility in different tasks.

The guidance mechanism during inference (Formula 4) combines the ideas of diffusion model denoising and gradient optimization. It is conceptually dense and may require readers to have a certain understanding of the relevant background to fully digest it.

---

> ### Author Rebuttal · Authors · 2025-07-31
>
> We thank Reviewer jkE6 for their time and expertise in reviewing our submission. Their insightful comments are valuable in helping us improve the quality of our work.
> > *“The method introduces some hyperparameters that need to be adjusted, such as the threshold and guidance strengths. How should $\tau$ be set in practice? Is it a sensitive parameter that needs to be fine-tuned for each task, or is it robust over a wide range? The sensitivity analysis of these hyperparameters is not sufficient in the main text of the paper, which may affect its reproducibility in different tasks.”*
>
> We thank the reviewer for pointing out this important issue. We agree that analyzing the hyperparameter design choice more explicitly in the main paper would be more informative for the readers, and **we will update the camera-ready version to include them.**
>
> We analyze the effect of varying the OOD threshold $\tau$ for the Transport and Tool-Hang tasks. The performance is robust to $\tau$ within a small range. We observe that setting $\tau$ too high results in fewer corrections being triggered, leading to performance drops. Conversely, setting $\tau$ too low can cause the dynamics model to get stuck in local optima and stall task progress. This effect is particularly pronounced in the Tool-Hang task, where the robot may get stuck at inserting the hook into the base.
>
> As described in the appendix, we select the threshold by first rolling out the base policy to identify the minimum OOD score associated with failure, then sweeping \tau below that value during evaluation with our method. The ablation experiment on guidance scale already in Appendix A.1. The optimal guidance scale is found by grid search.
>
> **Transport Task:**
>
> | $\\tau$ | Success Rate        |
> |------------------------|---------------------|
> | 1.6                    | 0.78 ± 0.028        |
> | 2.2                    | 0.82 ± 0.025        |
> | 2.8                    | 0.85 ± 0.009        |
> | 3.4                    | 0.76 ± 0.033        |
> | 4.0                    | 0.67 ± 0.038        |
>
> **Tool-Hang Task:**
>
> | $\\tau$ | Success Rate        |
> |------------------------|---------------------|
> | 0.8                    | 0.28 ± 0.028        |
> | 1.1                    | 0.33 ± 0.034        |
> | 1.4                    | 0.39 ± 0.009        |
> | 1.7                    | 0.34 ± 0.016        |
> | 2.0                    | 0.29 ± 0.034        |
>
> > *“On the quality of the latent space: The effectiveness of the entire LPB framework relies on a high-quality latent space. Can you discuss under what circumstances the visual encoder learned by the BC strategy may not form a "good" latent space (i.e., a space in which distances do not effectively reflect the degree of OOD)? For example, how does the performance of LPB change in the presence of a large amount of visual distractors that are irrelevant to the task.”*
>
> We thank the reviewer for this insightful question. Indeed, the effectiveness of LPB depends on a latent space where distances reflect task-relevant similarity. While learning the right representation for planning and control is still an open research question, our motivation for using the encoder trained with BC loss is that it encourages control-aware representations by aligning observations with expert actions. Below we discuss two scenarios in which the encoder learned in this way may sometimes struggle: (1) visually cluttered environments, where irrelevant features can have a dominating effect, and (2) limited visual diversity in demonstration, which can lead to overfitting and poor generalization.
>
> For (1), in our real-world setup, we use the pretrained in-the-wild policy encoder. The encoder and policy were trained on visually diverse images with task-irrelevant features. As shown in Fig. 7, the OOD score and nearest-neighbor matches are noisier in these settings, reflecting challenges in the latent space. For (2), in simulated tasks, where backgrounds are cleaner, we still observe occasional overfitting to visual features such as the robot’s appearance, which is present in the observation space.
>
> Future work on improving the latent space includes incorporating contrastive learning [1,2] or auxiliary objectives [3] that explicitly promote control-aware latent structure.
>
> > *“Comparison with explicit recovery strategies: Your method can be viewed as an online, implicit recovery strategy. Another idea is to explicitly train a separate recovery strategy using the same rollout data. Can you fundamentally analyze the advantages and disadvantages of your proposed "online guidance" method compared with the "offline training recovery strategy" method and what are the applicable scenarios?”*
>
> We thank the reviewer for this insightful question. Explicitly training a recovery policy from offline data is often challenging, as it typically requires access to task rewards [4] or prior assumptions about tasks such as equivariance [5]. In contrast, our proposed method does not require such additional assumptions. In addition, our method provides online correction through gradient-based action optimization guided by a learned dynamics model, conceptually similar to classifier guidance in diffusion models. This ensures that the corrected actions remain close to the expert manifold, unlike a separately trained recovery policy, which may drift outside the expert distribution and introduce instability.
>
> That said, offline recovery policies may be more sample-efficient when recovery behaviors are well-represented and consistent in the offline dataset. In comparison, LPB incurs higher test-time computational cost due to online optimization. We believe the two approaches are complementary: LPB offers adaptability and robustness to pre-trained base policies in novel online situations, while explicit recovery policies are well-suited for structured environments with known failure modes.
>
> ### References
>
> [1] Laskin, Michael, Aravind Srinivas, and Pieter Abbeel. *Curl: Contrastive unsupervised representations for reinforcement learning.* International conference on machine learning. PMLR, 2020.
>
> [2] Yan, Wilson, et al. *Learning predictive representations for deformable objects using contrastive estimation.* Conference on Robot Learning. PMLR, 2021.
>
> [3] Hansen, Nicklas, et al. *Self-supervised policy adaptation during deployment.* arXiv preprint arXiv:2007.04309 (2020).
>
> [4] Thananjeyan, Brijen, et al. *Recovery rl: Safe reinforcement learning with learned recovery zones.* IEEE Robotics and Automation Letters 6.3 (2021): 4915-4922.
>
> [5] Reichlin, Alfredo, et al. *Back to the manifold: Recovering from out-of-distribution states.* 2022 IEEE/RSJ International Conference on Intelligent Robots and Systems (IROS). IEEE, 2022.

---

> > ### Comment · Reviewer_jkE6 · 2025-08-07
> >
> > Thanks for the clarification, which addresses some of my concerns. I will keep my scores.

---

> ### Author Response · Authors · 2025-08-04
> **Looking forward to your feedback as the discussion phase draws to an end**
>
> Dear reviewer jkE6,
>
> We want to sincerely thank you again for your comments and suggestions, which has significantly improved the quality of our submission. We hope that our rebuttal responses have addressed your questions and concerns.
>
> As there are only two days left in the discussion phase, please don’t hesitate to let us know if you have any further questions or if we can provide additional clarifications to improve the score. We sincerely thank the reviewer for their time and effort in helping us improve our paper—your advice and feedback are greatly appreciated.
>
> Thanks,
>
> Authors

---

### Official Review · Reviewer_76MU · 2025-07-03

**Clarity:** 4
**Significance:** 3
**Originality:** 3
**Rating:** 5
**Confidence:** 4

**Summary:**

This paper presents a new approach to improve OOD recovery of the diffusion policy. The authors achieve this by learning a separate dynamics model and using classifier guidance to guide the generation process to in-distribution states to encourage recovery. The result is verified on well-designed experiments both in simulation and in the real world.

**Questions:**

1. How does the method compare to using a two-stage framework? For example, how about using planners such as MPPI with the learned dynamics model to push back into the expert distribution first, then utilize the expert diffusion model to do inference? Is this the MPC baseline in your appendix?
2. How much overhead (both memory for loading the expert dataset latents and computation for calling the learned dynamics model and calculating the gradient-based guidance) does the method add to the original diffusion policy inference pipeline?
3. When expert data size is scaled up, it seems both the proposed approach and the base expert policy lead to similar performance. I am curious what is the trend of the correction rate (i.e., percentage of times that the guidance is activated) as one increase the amount of expert data.

**Ethical Concerns:**

["NO or VERY MINOR ethics concerns only"]

**Final Justification:**

My final recommendation for this paper is 5 (Accept). The methods are well explained, and the experiments are well documented and demonstrate the good performance of their method in OoD recovery. The rebuttal also clarified several important questions I had on the baseline design and speed considerations. I am not fully convinced by their explanations on the claim that the method is fully plug-and-play. I think training a separate dynamis model is not a small task: it requires extra data gathering, network design, and hyperparameter tuning. Overall, I think the proposed method is a good improvement upon diffusion policies for OoD recovery.

**Limitations:**

Yes.

**Paper Formatting Concerns:**

None.

**Quality:**

3

**Strengths And Weaknesses:**

Strengths:

1. Presented a novel approach to recover from OOD situations in diffusion policies.
2. Well-designed experiments that showcase the effectiveness of their work.
3. Very effective improvements in the low (expert) data situations compared to previous works.
4. Overall clearly written and presented paper.

Weaknesses:

1. Classifier-based guidance in diffusion policy has been widely used already.
2. The proposed method claims to be Plug-And-Play. However, it still requires non-trivial data collection and training for the dynamics model.

---

> ### Author Rebuttal · Authors · 2025-07-30
>
> We thank Reviewer 76MU for their time and expertise in reviewing our submission. Their insightful comments are valuable in helping us improve the quality of our work.
> > *“How does the method compare to using a two-stage framework? For example, how about using planners such as MPPI with the learned dynamics model to push back into the expert distribution first, then utilize the expert diffusion model to do inference? Is this the MPC baseline in your appendix?”*
>
> We thank the reviewer for this important question. The MPC baseline in our appendix corresponds to a two-stage framework where action initializations proposed by the expert diffusion policy are refined using MPC with the learned dynamics model. We initially experimented with sampling actions from a Gaussian distribution for MPPI and MPC, but found that this leads to poor performance, as most samples fall out of distribution for both the dynamics model and the base policy, resulting in unreliable predictions. Using action proposals from the expert policy as initialization significantly improves stability and performance. While this MPC baseline does improve upon the base policy, we found it to be less effective than our gradient-based LPB correction, which offers more direct and fine-grained alignment with the expert manifold.
>
> > *“When expert data size is scaled up, it seems both the proposed approach and the base expert policy lead to similar performance. I am curious what is the trend of the correction rate (i.e., percentage of times that the guidance is activated) as one increase the amount of expert data.”*
>
> We thank the reviewer for this thoughtful question. Indeed, as the size of the expert dataset increases, the base policy becomes more reliable, and the benefit of LPB begins to saturate. Correspondingly, we observe that the frequency of invoking gradient-guided correction decreases with more expert data. Below, we report the correction rate across different fractions of the demonstration set. As the coverage of demonstration data grows, the latent OOD score will decrease, causing the correction to be invoked less often. LPB is most beneficial in low-data regimes and naturally defers to the base policy when sufficient coverage is achieved. **We will include this discussion in the final paper**.
>
> **Tool-Hang Task:**
> | Demo Data Perc.      | 20%  | 40%  | 60%  | 80%  | 100% |
> |----------------|------|------|------|------|-------|
> | Correction Perc. | 77%  | 62%  | 55%  | 39%  | 35%  |
>
> > *“How much overhead (both memory for loading the expert dataset latents and computation for calling the learned dynamics model and calculating the gradient-based guidance) does the method add to the original diffusion policy inference pipeline?”*
>
> We thank the reviewer for this important question. Below we provide detailed measurements of the real-time computational cost of LPB's gradient-based steering during inference on the real world cup arrangement task. **We will update the paper to make this more clear.**
>
> We predict 16 action steps and execute the first 12. On a GeForce RTX 3080 GPU, it takes around 5 minutes to load the expert dataset and caching them to latent representations. The base policy has an initial overhead of 0.36 seconds and 2.0 GB GPU usage. After warm-up, inference runs at **77 ms** per chunk with peak usage of **1.6 GB**. LPB incurs an initial overhead of 0.87 seconds and 3.3 GB usage, and after warm-up, inference runs at **420 ms** per chunk with peak memory usage of **3.0 GB**.
>
> > *“The proposed method claims to be Plug-And-Play. However, it still requires non-trivial data collection and training for the dynamics model.”*
>
> We thank the reviewer for this insightful comment. We fully acknowledge that training a dynamics model requires additional rollout data, which can be non-trivial in real-world settings. However, we emphasize that our method remains plug-and-play in the sense that it does not require task-specific supervision, reward engineering, or labeled failure cases, only passive rollout data, which can be collected in an unsupervised manner. This makes LPB broadly applicable and easy to integrate in practice.

---

> > ### Comment · Reviewer_76MU · 2025-08-05
> >
> > Thank you for the clarification. This addressed all of my questions. I will maintain my score.

---

> ### Author Response · Authors · 2025-08-04
> **Looking forward to your feedback as the discussion phase draws to an end**
>
> Dear reviewer 76MU,
>
> We want to sincerely thank you again for your comments and suggestions, which has significantly improved the quality of our submission. We hope that our rebuttal responses have addressed your questions and concerns.
>
> As there is only 2 days left in the discussion phase, should you have any further questions, please don't hesitate to let us know and we'll be happy to address them.
>
> Thanks,
>
> Authors

---

### Official Review · Reviewer_9nMP · 2025-07-14

**Clarity:** 3
**Significance:** 3
**Originality:** 4
**Rating:** 5
**Confidence:** 4

**Summary:**

This paper introduces Latent Policy Barrier (LPB), a novel framework for robust visuomotor policy learning via behavior cloning. LPB decouples precise expert imitation from out-of-distribution (OOD) recovery by combining: (1) a base diffusion policy trained exclusively on high-quality expert demonstrations to ensure task performance, and (2) a visual latent dynamics model trained on both expert and automatically collected rollout data to detect and correct deviations at inference time. At test time, LPB computes a latent OOD score-the $l_2$ distance between the current latent embedding and its nearest expert neighbor-and, if it exceeds a threshold, performs gradient-based steering in latent space to pull predicted future states back toward the expert manifold. Experiments on four simulated robotic manipulation tasks (Push-T, Square, Tool-Hang, Transport) and a real-world cup-arrangement task demonstrate that LPB achieves up to a 34% success-rate improvement over strong baselines under limited expert data and maintains robustness under action noise and significant distribution shifts.

**Questions:**

1. How sensitive is LPB's performance to the choice of latent-OOD threshold $\tau$? Can you provide guidance or automate its selection?

2. What is the real-time computational cost of LPB's gradient-based steering during inference, and how might this scale with higher action dimensions or longer denoising chains?

3. Storing all expert latent embeddings for nearest-neighbor search can be memory-intensive. Have you considered approximate methods (e.g., k-d trees or learned quantization) and their impact on robustness?

4. LPB focuses on short-term, local corrections. How might the dynamics model be extended for long-horizon planning to recover from more severe drifts?

**Ethical Concerns:**

["NO or VERY MINOR ethics concerns only"]

**Final Justification:**

Rebuttal of the authors (https://openreview.net/forum?id=FUd016XD4d&noteId=VuhVVwxj5q) has address all my concerns about this work. (Thanks for your detailed clarification. Good work!) I decided to change my rating to 5: Accept.

**Limitations:**

Yes.

**Paper Formatting Concerns:**

None.

**Quality:**

4

**Strengths And Weaknesses:**

# Quality

**Strengths**: the empirical evaluation is thorough, covering both simulated and real-robot benchmarks. LPB consistently outperforms baselines-including Behavior Cloning variants, Filtered BC, CCIL, and CQL-especially in low-data regimes (20% demos) and under inference-time perturbations (up to 0.4 action noise).

**Weaknesses**: the paper lacks theoretical guarantees on convergence or safety margins. Evaluation is limited to a narrow set of manipulation tasks; generalization to other domains (e.g., mobile navigation) remains untested.

# Clarity

**Strengths**: the conceptual analogy to Control Barrier Functions and the separation of training vs. inference components are well motivated. Figures (e.g., rollout comparisons in Figure 5) and Algorithm 1 clearly illustrate the LPB pipeline.

**Weaknesses**: Key hyperparameters, such as the OOD threshold $\tau$, guidance scale $\eta$, and selection of nearest-neighbor search strategy, are deferred to the appendix, which may hinder immediate reproducibility. The presentation of computational overhead at inference (gradient-guided denoising) could be more detailed.

# Significance

**Strengths**: Covariate shift is a central challenge in imitation-based visuomotor control. LPB offers a plug-and-play approach that enhances robustness without additional human labeling or policy fine-tuning, potentially benefiting a broad class of pretrained vision-based policies.

**Weaknesses**: While impactful within robotic manipulation, the approach's scalability to environments with very high-dimensional observation spaces or multi-agent settings is unclear.

# Originality

**Strengths**: LPB is the first to frame expert demonstrations as an implicit latent barrier and to perform gradient-based inference steering in a learned visual latent space for behavior cloning policies. This unifies ideas from diffusion-policy classifier guidance and model-predictive refinement in a novel way.

**Weaknesses**: Related works on guided denoising and model-based policy refinement exist; the incremental novelty lies primarily in applying these to latent-space neighbor distances rather than explicit reward signals.

---

> ### Author Rebuttal · Authors · 2025-07-30
>
> We thank Reviewer 9nMP for their time and expertise in reviewing our submission. Their insightful comments are valuable in helping us improve the quality of our work.
> > *“Evaluation is limited to a narrow set of manipulation tasks; generalization to other domains (e.g., mobile navigation) remains untested.”*
>
> We thank the reviewer for this comment. Our work focuses on long-horizon, precise manipulation tasks where covariate shifts can cause unrecoverable failures. To further test generalization within this domain, we evaluate on Libero-10 [1], a multi-task, language-conditioned manipulation benchmark that contains 10 different tasks. For each task, Libero-10 provides 50 expert demonstration trajectories. We use those $50\times 10$ expert trajectories to train the base policy. For each task, we collect 200 rollout trajectories. We use $200\times 10$ rollout trajectories and the 500 expert trajectories to train the dynamics model. Both the base policy and the dynamics model are multi-task with an additional shared language encoder. We perform 50 rollouts per task across ten tasks and report the average success rate.
>
> | Method        | Libero-10 Success Rate |
> |---------------|------------------------|
> | Expert BC     | 0.65 ± 0.009           |
> | Mixed BC      | 0.50 ± 0.017           |
> | Filtered BC   | 0.71 ± 0.006           |
> | CCIL          | 0.61 ± 0.026           |
> | Ours (LPB)    | **0.75 ± 0.038**       |
>
> We also conduct real-world evaluation on the **belt assembly task** from the NIST Task Board. We collect 200 expert trajectories to train the base policy and 400 rollout trajectories to train the dynamics model. For both LPB and the base policy, we perform 40 rollouts. LPB improves the success rate from **0.55** to **0.75**. Please see the project website for rollouts.
>
>
> > *“Key hyperparameters, such as the OOD threshold, guidance scale, and selection of nearest-neighbor search strategy, are deferred to the appendix, which may hinder immediate reproducibility. How sensitive is LPB's performance to the choice of latent-OOD threshold? Can you provide guidance or automate its selection?”*
>
> We thank the reviewer for pointing out this presentation issue. We agree that presenting the hyperparameter design choice more explicitly in the main paper would be more informative for the readers, and **we will include them in the camera-ready version**.
>
> We analyze the effect of varying the OOD threshold $\\tau$ for the Transport and Tool-Hang tasks. We observe that setting $\\tau$ too high results in fewer corrections being triggered, leading to performance drops. Conversely, setting $\\tau$ too low can cause the dynamics model to get stuck in local optima and stall task progress. This effect is particularly pronounced in the Tool-Hang task, where the robot may get stuck at inserting the hook into the base.
>
> As described in the appendix, we select the threshold by first rolling out the base policy to identify the minimum OOD score associated with failure, then sweeping $\\tau$ below that value during evaluation with our method.
>
> **Transport Task:**
>
> | $\\tau$ | Success Rate        |
> |------------------------|---------------------|
> | 1.6                    | 0.78 ± 0.028        |
> | 2.2                    | 0.82 ± 0.025        |
> | 2.8                    | 0.85 ± 0.009        |
> | 3.4                    | 0.76 ± 0.033        |
> | 4.0                    | 0.67 ± 0.038        |
>
> **Tool-Hang Task:**
>
> | $\\tau$ | Success Rate        |
> |------------------------|---------------------|
> | 0.8                    | 0.28 ± 0.028        |
> | 1.1                    | 0.33 ± 0.034        |
> | 1.4                    | 0.39 ± 0.009        |
> | 1.7                    | 0.34 ± 0.016        |
> | 2.0                    | 0.29 ± 0.034        |
>
> > *“What is the real-time computational cost of LPB's gradient-based steering during inference, and how might this scale with higher action dimensions or longer denoising chains?”*
>
> We thank the reviewer for this important question. Below we provide detailed measurements of the real-time computational cost of LPB's gradient-based steering during inference on the real world cup arrangement task. **We will update the paper to make this more clear.**
>
> We predict 16 action steps and execute the first 12. On a GeForce RTX 3080 GPU, it takes around 5 minutes to load the expert dataset and caching them to latent representations. The base policy has an initial overhead of 0.36 seconds and 2.0 GB GPU usage. After warm-up, inference runs at **77 ms** per chunk with peak usage of **1.6 GB**. LPB incurs an initial overhead of 0.87 seconds and 3.3 GB usage, and after warm-up, inference runs at **420 ms** per chunk with peak memory usage of **3.0 GB**.
>
> In summary, LPB’s inference cost scales reasonably with an action dimension of 10 (3D translation, 6D rotation, and 1D gripper control) and a horizon of 16. For quasi-static tasks, this cost is not a bottleneck in practice. However, the computational cost would increase with longer denoising chains due to additional gradient steps during backpropagation.
>
> > *"Storing all expert latent embeddings for nearest-neighbor search can be memory-intensive. Have you considered approximate methods (e.g., k-d trees or learned quantization) and their impact on robustness?"*
>
> We thank the reviewer for the insightful question. Indeed, storing all expert latents can be memory intensive. In this work, the out-of-distribution detection is performed in the most straightforward way - latent l2 distance to the nearest expert latent. Exploring more scalable alternatives, such as approximate nearest-neighbor search, is definitely a promising direction that could reduce memory and compute overhead.
>
> > *"LPB focuses on short-term, local corrections. How might the dynamics model be extended for long-horizon planning to recover from more severe drifts?"*
>
> We thank the reviewer for the insightful question. As noted, LPB is most effective for short-term, local corrections. Extending it to longer horizons would require a dynamics model with stronger generalization and broader data coverage. In principle, sampling-based approaches combined with a reliable dynamics model can support long-horizon recovery [2,3], and integrating such long-term planning capabilities is an interesting direction for future work.
>
> > *"the paper lacks theoretical guarantees on convergence or safety margins"*
>
> We thank the reviewer for the thoughtful question. We fully acknowledge that our paper does not provide theoretical guarantees. Providing theoretical guarantees for learning based systems is still an open challenge [4,5] and is an exciting future direction to our paper. Despite lack of theoretical guarantees, our extensive experiments have shown the practical value of gradient-based correction with a dynamics model.
>
> > *"While impactful within robotic manipulation, the approach's scalability to environments with very high-dimensional observation spaces or multi-agent settings is unclear."*
>
> We thank the reviewer for this thoughtful comment. Our method is explicitly designed for high-dimensional observation spaces and all tasks in our evaluation rely on raw RGB images (some with more than one camera view) as input. We demonstrate strong empirical performance across a diverse set of vision-based, long-horizon manipulation tasks, indicating that our approach scales well in high-dimensional observation domains.
>
> Regarding multi-agent settings, our method is not directly formulated to handle joint policies or shared dynamics between agents. LPB assumes a single-agent setting where the dynamics model and OOD guidance operate on individual observations and actions. Extending this framework to multi-agent scenarios is an exciting future direction and would require rethinking several core components, such as modeling joint dynamics, defining agent-specific guidance objectives, and addressing non-stationarity introduced by other agents. We view this as a non-trivial and interesting research direction but leave it outside the scope of the current work.
>
>
> ### References
> [1] Liu, Bo, et al. *Libero: Benchmarking knowledge transfer for lifelong robot learning.* Advances in Neural Information Processing Systems 36 (2023): 44776-44791.
>
> [2] Feng, Yunhai, et al. *Finetuning offline world models in the real world.* arXiv preprint arXiv:2310.16029 (2023).
>
> [3] Zhou, Gaoyue, et al. *Dino-wm: World models on pre-trained visual features enable zero-shot planning.* arXiv preprint arXiv:2411.04983 (2024).
>
> [4] Dixit, Anushri, et al. *Perceive with confidence: Statistical safety assurances for navigation with learning-based perception.* 8th Annual Conference on Robot Learning. 2024.
>
> [5] Majumdar, Anirudha, Alec Farid, and Anoopkumar Sonar. *PAC-Bayes control: learning policies that provably generalize to novel environments.* The International Journal of Robotics Research 40.2–3 (2021): 574–593.

---

> ### Author Response · Authors · 2025-08-04
> **Looking forward to your feedback as the discussion phase draws to an end**
>
> Dear reviewer 9nMP,
>
> We want to sincerely thank you again for your comments and suggestions, which has significantly improved the quality of our submission. We hope that our rebuttal responses have addressed your questions and concerns.
>
> As there are only two days left in the discussion phase, please don’t hesitate to let us know if you have any further questions or if we can provide additional clarifications to improve the score. We sincerely thank the reviewer for their time and effort in helping us improve our paper, your advice and feedback are greatly appreciated.
>
> Thanks,
>
> Authors

---

> > ### Comment · Reviewer_9nMP · 2025-08-07
> >
> > The detailed rebuttal has address all my concerns about this work. I have changed my score to 5: Accept. Good work!

---

### Author Response · Authors · 2025-08-06
**Thank you and we are looking forward to your post-rebuttal feedback!**

Dear AC and all reviewers:

We want to thank all reviewers for your valuable feedback, which has greatly improved the clarity and quality of our submission. We have incorporated additional experiment results and clarifications in our rebuttal response. As the discussion phase draws to a close, we look forward to hearing your post-rebuttal responses. If you have any further suggestions, be it additional experiments or points of clarification, please let us know. Your insights are deeply appreciated!

Thanks,
Authors

---

### Note · Authors · 2025-08-16

Dear Area Chairs and Reviewers,

We sincerely thank the reviewers for their time, insightful comments, and constructive feedback throughout the review process. We greatly appreciate the recognition of our submission. In our rebuttal and discussion, we have addressed all raised questions and provided additional experiments and discussion to further strengthen our submission:

- **More diverse and challenging tasks**
  - *Libero-10 (simulation):* Demonstrated that our framework can work on multi-task, language-conditioned, long-horizon manipulation.
  - *Belt assembly (real robot):* Showed improved robustness in contact-rich, real-world manipulation.

- **Hyperparameter choices and computational cost**
  - Added an ablation study on the OOD score threshold.
  - Explained design rationales behind hyperparameter selections.
  - Reported the computational cost of running LPB on a real robot.

- **Clarifications & Discussion**
  - Clarified differences between LPB and two-stage recovery-policy frameworks.
  - Discussed the quality of the learned latent space.
  - Discussed alternative uncertainty quantification methods.
  - Discussed the trend of correction rate.


We thank the reviewers again for their valuable feedback. We will incorporate these insights, additional experimental results, and discussions into our final revision.

---

### Decision · Program_Chairs · 2025-09-17

**Decision:**

Accept (spotlight)

**Comment:**

The paper proposes a novel framework to enhance behavior cloning diffusion policy, by learning a separate dynamics model and using classifier guidance to guide the generation process to in-distribution states to encourage recovery. In the discussion stage the reviewers are convinced by further experiments on additional environments and sensitivity to hyper-parameters. They are in consensus that this paper should be accepted.